# Multiplexed storage and real-time manipulation based on a multiple degree-of-freedom quantum memory

Tian-Shu Yang[1,2], Zong-Quan Zhou[1,2], Yi-Lin Hua[1,2], Xiao Liu[1,2], Zong-Feng Li[1,2], Pei-Yun Li[1,2], Yu Ma[1,2], Chao Liu[1,2], Peng-Jun Liang[1,2], Xue Li[1,2], Yi-Xin Xiao[1,2], Jun Hu[1,2], Chuan-Feng Li[1,2] & Guang-Can Guo[1,2]

The faithful storage and coherent manipulation of quantum states with matter-systems would enable the realization of large-scale quantum networks based on quantum repeaters. To achieve useful communication rates, highly multimode quantum memories are required to construct a multiplexed quantum repeater. Here, we present a demonstration of on-demand storage of orbital-angular-momentum states with weak coherent pulses at the single-photon-level in a rare-earth-ion-doped crystal. Through the combination of this spatial degree-of-freedom (DOF) with temporal and spectral degrees of freedom, we create a multiple-DOF memory with high multimode capacity. This device can serve as a quantum mode converter with high fidelity, which is a fundamental requirement for the construction of a multiplexed quantum repeater. This device further enables essentially arbitrary spectral and temporal manipulations of spatial-qutrit-encoded photonic pulses in real time. Therefore, the developed quantum memory can serve as a building block for scalable photonic quantum information processing architectures.

[1] CAS Key Laboratory of Quantum Information, University of Science and Technology of China, Hefei 230026, China. [2] Synergetic Innovation Center of Quantum Information and Quantum Physics, University of Science and Technology of China, Hefei 230026, P.R. China. Correspondence and requests for materials should be addressed to Z.-Q.Z. (email: zq_zhou@ustc.edu.cn) or to C.-F.L. (email: cfli@ustc.edu.cn)

L arge-scale quantum networks would enable long-distance quantum communication and optical quantum computing[1–3]. Due to the exponential photon loss in optical fibers[4], quantum communication via ground-based optical fibers is currently limited to distances of hundred of kilometers. To overcome this problem, the idea of quantum repeater[5,6] has been proposed to establish quantum entanglement over long distances based on quantum memories and entanglement swapping. It has been shown that to reach practical data rates using this approach, the most significant improvements can be achieved through the use of multiplexed quantum memories[7–9].

The multiplexing of quantum memories can be implemented using any degree-of-freedom (DOF) of the photons, such as those in the temporal[10], spectral[11], and spatial[12] domains. Rare-earth-ion-doped crystals (REIC) offer interesting possibilities as multiple-DOF quantum memories for photons by virtue of their large inhomogeneous bandwidths[11,13,14] and long coherence time[15] at cryogenic temperatures. Recently, there have been several important demonstrations using REIC, such as the simultaneous storage of 100 temporal modes[16] by atomic frequency comb (AFC)[17] featuring preprogrammed delays, the storage of tens of temporal modes by spin-wave AFC with on-demand readout[18–22] and the storage of 26 frequency modes with feed-forward controlled readout[11]. The orbital-angular-momentum (OAM) of a photon receives much attention because of the high capacity of OAM states for information transmission and spatial multimode operations[23]. Tremendous developments have recently been achieved in quantum memories for OAM states[24–26], paving the way to quantum networks and scalable communication architectures based on this DOF.

To date, most experiments with quantum memories have been confined to the storage of multiple modes using only one DOF, e.g., temporal, spectral, or spatial. To significantly improve the communication capacity of quantum memories and quantum channels, we consider a quantum memory using more than one DOF simultaneously[11,27–29].

Here, we report on the experimental realization of an on-demand quantum memory storing single photons encoded with three-dimensional OAM states in a REIC. We present the results of a multiplexed spin-wave memory operating simultaneously in temporal, spectral and spatial DOF. In addition to expanding the number of modes in the memory through parallel multiplexing, a quantum mode converter (QMC)[30] can also be realized that can perform mode conversion in the temporal and spectral domains simultaneously and independently. Indeed, our quantum memory enables arbitrary temporal and spectral manipulations of spatial-qutrit-encoded photonic pulses, and thus can serve as a real-time sequencer[14], a real-time multiplexer/demultiplexer[31], a real-time beam splitter[32], a real-time frequency shifter[33], a real-time temporal/spectral filter[31], among other functionalities.

## Results

### Experimental setup.
A scheme of our experimental setup and relevant atomic level structure of $Pr^{3+}$ ions is presented in Fig. 1. The memory crystal (MC) and filter crystal (FC) used in this setup are $3 \times 6 \times 3$ mm crystals of 0.05% doped $Pr^{3+}$:$Y_2SiO_5$, which are cooled to 3.2 K using a cryogen-free cryostat (Montana Instruments Cryostation). In order to maximize absorption, the polarization of input light is close to the $D_2$-axis of $Y_2SiO_5$ crystal. To realize reliable quantum storage with high multimode capacity, we created a high-contrast AFC in MC (the AFC structure is shown in Supplementary Note 1). Spin-wave storage is employed to enable on-demand retrieval and extend storage time[17]. The control and input light are steered towards the MC in opposite directions with an angular offset ~4° to reject the strong control field and avoid the detection of free induction decay noise[34]. To achieve a low noise floor, we increase the absorption depth of the FC by employing a double-pass configuration. Figure 2a presents

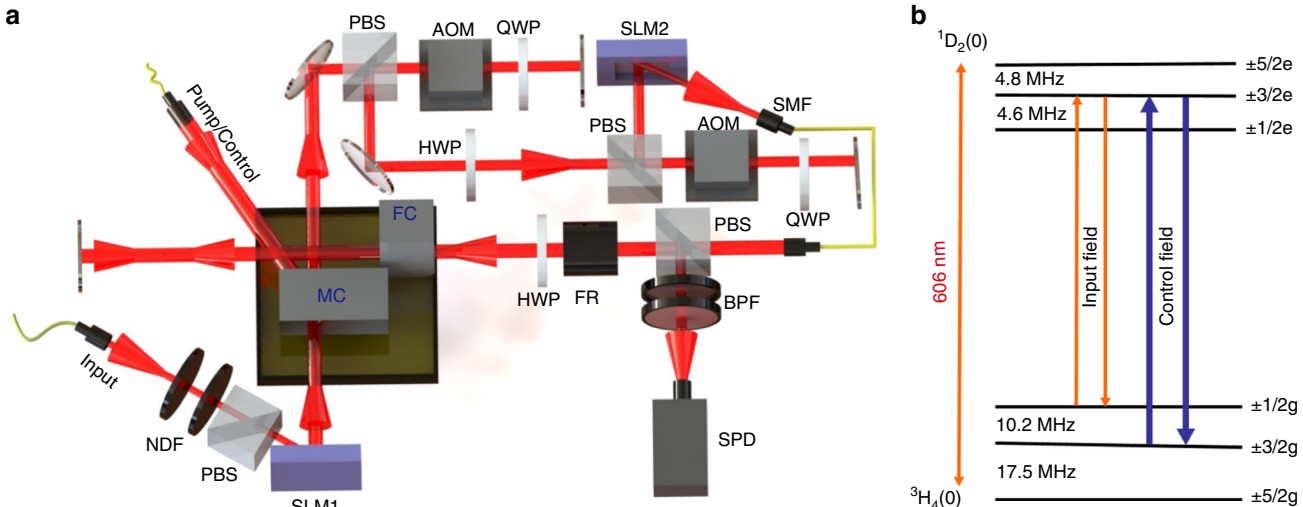

**Fig. 1** Experimental setup and atomic levels. **a** Schematic illustrations of the experiment. The AFC is prepared in a memory crystal (MC) and a narrow spectral filter has been prepared in a filter crystal (FC)[49]. The beam waist of the input light is 65 μm at the middle of the MC. The pump/control light has a beam waist of 300 μm inside the MC to ensure good overlap with the high-dimensional input light. The input pulses are attenuated to single-photon level by neutral density filters (NDF). The spatial modes of these photons are converted into OAM superposition states by a spatial light modulator (SLM1). After storage in the MC, the retrieved signal passes through two consecutive acousto-optical modulators (AOM)[42], which act as a temporal gate and a frequency shifter. The AOM are used in double-pass configuration to ensure the photons' spatial mode unchanged when the frequency of photons is swept over tens of MHz. The SLM2 and a single-mode fiber (SMF) are employed to analyze the OAM states of the retrieved photons. The FC is double-passed with help of a polarization beam splitter (PBS), a half-wave plate (HWP) and a Faraday rotator (FR). Two bandpass filters (BPF) centered at 606 nm are employed to further suppress noises before the final detection of signal photons using single-photon detector (SPD). QWP quarter-wave plate. **b** Hyperfine states of the first sublevels of the ground and the excited states of $Pr^{3+}$ in $Y_2SiO_5$. The input field is resonant with 1/2g–3/2e, and the control field is resonant with 3/2g–3/2e (see Methods section for details)

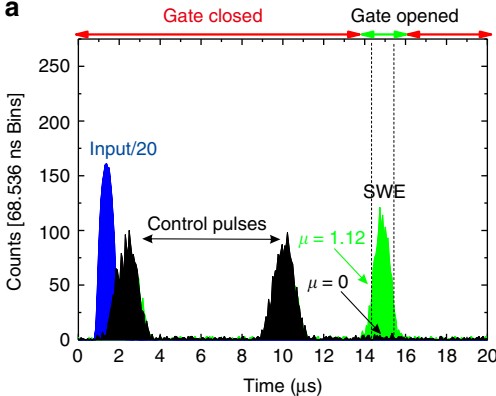

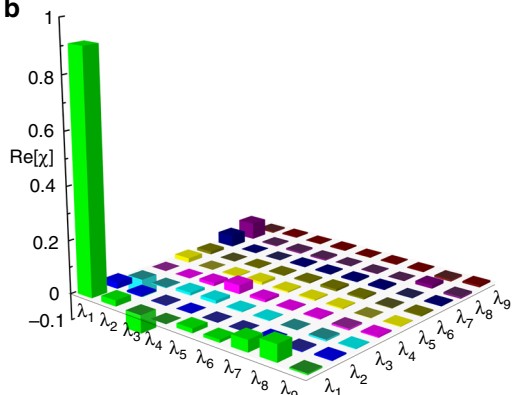

**Fig. 2** Time histograms and the reconstructed process matrix of quantum storage process in three-dimensional OAM space. **a** Time histograms of the input photons (blue), the photons retrieved at 12.68 μs for $\mu = 1.12$ (green) and the unconditional noise for $\mu = 0$ (black). **b** Graphical representation of the process matrix $\chi$ of memory process as estimated via quantum process tomography. Details of operators $\lambda_i$ are shown in Section Methods. Only the real part of the experimentally reconstructed process matrix is shown. All values are in the imaginary part are smaller than 0.090

the time histograms of the input photons (blue) and the photons retrieved at 12.68 μs (green) with a spin-wave storage efficiency $\eta_{SW} = 5.51\%$. For an input with a mean photon number $\mu = 1.12$ per pulse, we have measured a signal-to-noise ratio (SNR) ~39.7 ± 6.7 with the input photons in the Gaussian mode.

**Quantum process tomography**. The ability to realize the on-demand storage of photonic OAM superposition states in solid-state systems is crucial for the construction of OAM-based high-dimensional quantum networks[24]. Quantum process tomography for qutrit operations[35] benchmarks the storage performance for OAM qutrit in our solid-state quantum memory. The qutrit states are prepared in the following basis of OAM states: $\left| LG_{p=0}^{l=-1} \right\rangle$, $\left| LG_{p=0}^{l=0} \right\rangle$, $\left| LG_{p=0}^{l=1} \right\rangle$. Here, $\left| LG_{p}^{l} \right\rangle$ corresponds to OAM states defined as Laguerre-Gaussian (LG) modes, where $l$ and $p$ are the azimuthal and radial indices, respectively. In the following, we use the kets $|L\rangle$, $|G\rangle$, and $|R\rangle$ to denote the OAM states $\left| LG_{p=0}^{l=1} \right\rangle$, $\left| LG_{p=0}^{l=0} \right\rangle$, and $\left| LG_{p=0}^{l=-1} \right\rangle$, respectively. For $\mu = 1.12$, we first characterized the input states before the quantum memory using quantum state tomography (see Methods section for details). The reconstructed density matrices of input are not ideal because of imperfect preparation and measurements based on spatial light

modulators and single-mode fibers[26]. We then characterized the memory operation using quantum process tomography. Figure 2b presents the real part of the experimentally reconstructed process matrix $\chi$. It is found to have a fidelity of 0.909 ± 0.010 with respect to the Identity operation. This fidelity exceeds the classical bound of 0.831 (see Supplementary Note 4 for details), thereby confirming the quantum nature of the memory operation. The nonunity value of the memory fidelity may be caused by the limited beam waist of the pump/control light, which may result in imperfect overlap with different OAM modes. Moreover, we noted that the memory performance for superposition states of |L⟩ and |R⟩ is much better than that achieved here (as detailed in Supplementary Note 2). The visibility of such two-dimensional superposition states is higher than the fidelity of the memory process in all three dimensions. This result indicates that the storage efficiency is balanced for the symmetrical LG modes but is not balanced for all the three considered spatial modes.

**Multiplexing storage in multiple DOF**. Carrying information in multiple DOF on photons can expand the channel capacity of quantum communication protocols[11,36]. Here, we show that our solid-state memory can be simultaneously multiplexed in temporal, spectral, and spatial DOF. As shown in Fig. 3a, two AFC are created in the MC with an interval of 80 MHz between them to achieve spectral multiplexing. The two AFC have the same peak spacing of $\Delta = 200$ kHz and the same bandwidth of $\Gamma_{AFC} = 2$ MHz. The spin-wave storage efficiencies are 5.05% and 5.13%, for the first AFC and the second AFC, respectively. The temporal multimode capacity of an AFC is limited by $\Gamma_{AFC}/\Delta$[17]. However, increasing the number of modes, the time interval between the last control pulse and the first output signal pulse will be reduced. Therefore, we employed only two temporal modes to reduce the noise caused by the last control pulse. The spatial multiplexing is realized by using three independent paths as input as shown in Fig. 3b. These paths, $s_1$, $s_2$, and $s_3$, correspond to the OAM states as $|L\rangle$, $|R\rangle$, and $|G\rangle$ defined above. By combining all three DOF together, we obtain $2 \times 2 \times 3 = 12$ modes in total. The FC is employed to select out the desired spectral modes. Figure 3c illustrates the results of multimode storage over these three DOF for $\mu = 1.04$. The minimum crosstalk as obtained from mode crosstalk for each mode is 19.7 ± 3.41, which is calculated as one takes the counts in the diagonal term as the signal and then locates the large peaks over the range of output modes as the noise.

Here, the temporal, spectral and spatial DOF are employed as classical DOF for multiplexing. One can choose any DOF to carry quantum information. As a typical example, now we use the temporal and spectral DOF for multiplexing and each channel is encoded with spatial-qutrit state of $|\psi_1\rangle = (|L\rangle + |G\rangle + |R\rangle)/\sqrt{3}$. Each channel is labeled as $f_i t_j$, where $f_i$ represent spectral modes $i$ and $t_j$ similarly represent temporal modes $j$. Figure 4a shows the experimental results for $\mu = 1.04$. The minimum crosstalk as obtained from the mode crosstalk is approximately 15.2. We measured the fidelities of the spatial-qutrit state for each channel as shown in Fig. 4b.

A QMC can transfer photonic pulses to a target temporal or spectral mode without distorting the photonic quantum states. A real-time QMC that can operate on many DOF is essential for linking the components of a quantum network[30,37]. By adjusting the timing of the control pulse, one can specify the recall time in an on-demand manner to realize the temporal mode conversion. The two-AOM gate in our system can be used as a high-speed frequency shifter by tuning its driving frequency. Therefore, spectral and temporal mode conversion can be realized independently and simultaneously. Figure 4c presents the results of QMC operation for $\mu = 1.04$. We can convert from $f_i$ to $f_j$ and

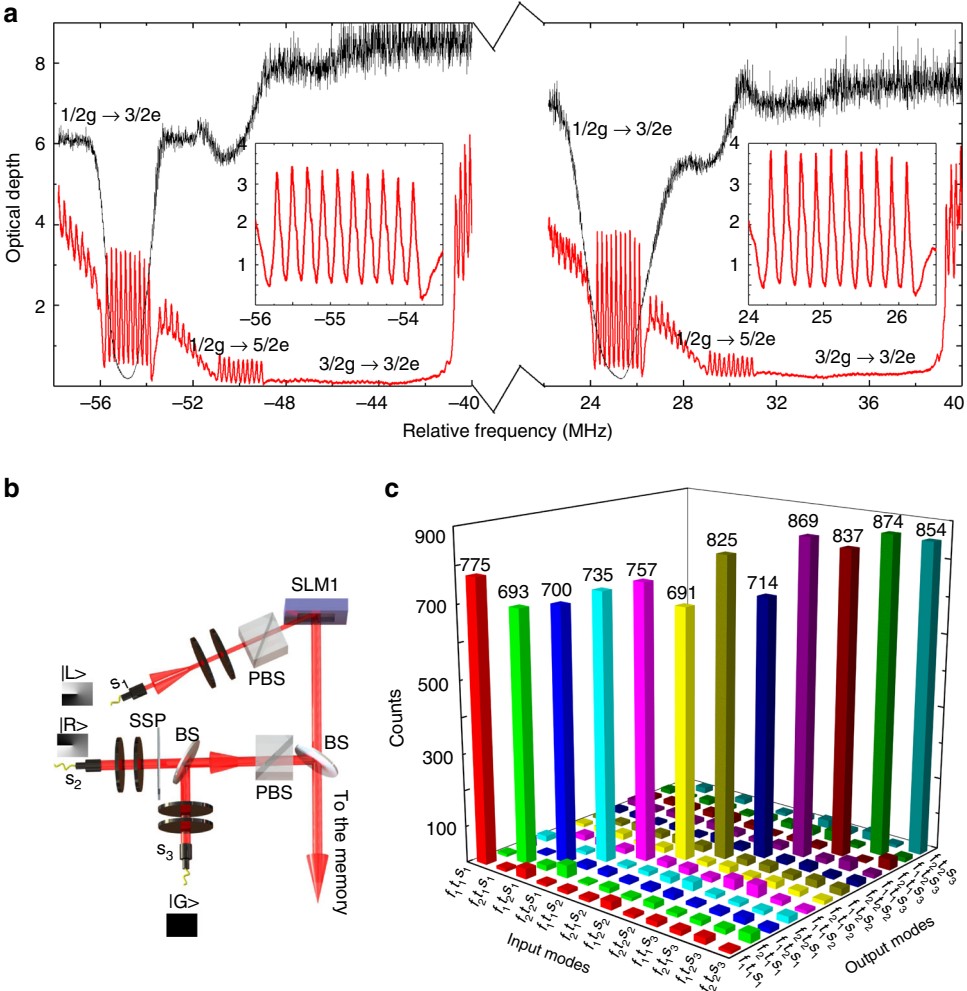

**Fig. 3** Multiplexed spin-wave storage in three degrees of freedom at the single-photon level. **a** The double AFC structure (red) in the MC and the double filter structure (black) in the FC. **b** Three independent spatial modes carrying different OAM states are employed for spatial multiplexing. The spatial mode $s_1$ is converted into |L⟩ state in SLM1. The spatial mode $s_2$ is converted into |R⟩ state in a spiral phase plate (SSP); the spatial mode $s_3$ is a Gaussian mode which is used as |G⟩ state. They are combined by two pellicle beam splitters (BS). **c** A demonstration of temporal, spectral and spatial multiplexed storage for single-photon level inputs

from $t_i$ to $t_j$, where these notations represent all different spectral modes and temporal modes. The noise level is significantly weaker than the strength of the converted signal, which indicates that the QMC operates quietly enough to avoid introducing any mode crosstalk. All these modes are encoded with OAM spatial-qutrit of $|\psi_1\rangle$. To demonstrate that the qutrit state coherence is well preserved after QMC operation, we measured the fidelities (see Methods for details) between the input and converted states. The results, presented in Fig. 4d, indicate that the QMC can convert arbitrary temporal and spectral modes in real-time while preserving their quantum properties. Our device is expected to find applications in quantum networks comprising two quantum memories, in which mismatched spectral or temporal photon modes may need to be converted[30]. This device can ensure the indistinguishability of the photons which are retrieved from any quantum memory. This device could find application in many photonic information processing protocols, e.g., a Bell-state measurement[11], and quantum memory-assisted multiphoton generation[9].

**Arbitrary manipulations in real time**. The precise and arbitrary manipulation of photonic pulses while preserving photonic coherence is an important requirement for many proposed

photonic technologies[31]. In addition to the QMC functionality demonstrated above, the developed quantum memory can enable arbitrary manipulations of photonic pulses in the temporal and spectral domains in real time. As an example, we prepared the OAM qutrit state $|\psi_1\rangle$ in the $f_1t_1$ and $f_2t_2$ modes (Fig. 5a) as the input. Four typical operations were demonstrated, i.e., exchange of the readout times for the $f_1$ and $f_2$ photons, the simultaneous retrieval of the $f_1$ and $f_2$ photons at $t_1$, shifting the frequency of $f_1$ photons to $f_2$ but keeping the frequency of $f_2$ photons unchanged and temporal beam splitting the $f_1$ photons but filtering out the $f_2$ photons. These operations correspond to output of $|\psi_1\rangle_{f_1t_2,f_2t_1}$, $|\psi_1\rangle_{f_1t_1,f_2t_2}$, $|\psi_1\rangle_{f_2t_1,f_2t_2}$, and $|\psi_1\rangle_{f_1t_1,f_1t_2}$, respectively. Another example was implemented with the OAM qutrit state $|\psi_2\rangle = (|L\rangle + |G\rangle - i|R\rangle)/\sqrt{3}$ encoded in the $f_1t_2$ and $f_2t_2$ modes as the input, as shown in Fig. 5b with same output. The retrieved states were then characterized via quantum state tomography as usual (see Methods). Table 1 shows the fidelities between output states and input states.

**Discussion**

In conclusion, we have experimentally demonstrated a multiplexed solid-state quantum memory that operates simultaneously

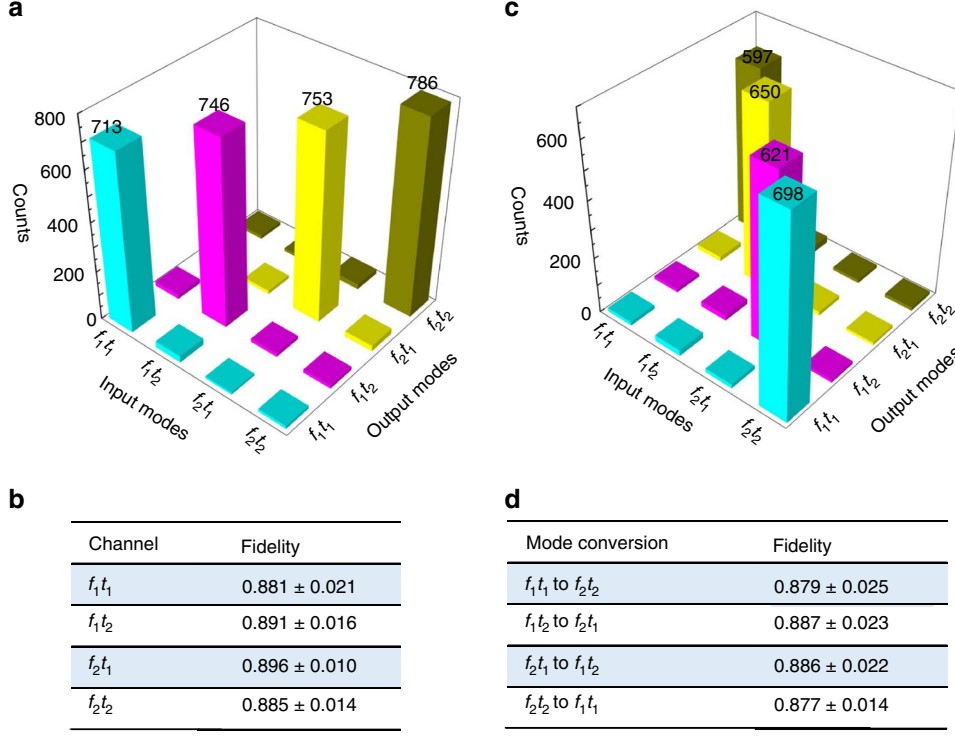

| Channel | Fidelity |
|---|---|
| $f_1 t_1$ | $0.881 \pm 0.021$ |
| $f_1 t_2$ | $0.891 \pm 0.016$ |
| $f_2 t_1$ | $0.896 \pm 0.010$ |
| $f_2 t_2$ | $0.885 \pm 0.014$ |

| Mode conversion | Fidelity |
|---|---|
| $f_1 t_1$ to $f_2 t_2$ | $0.879 \pm 0.025$ |
| $f_1 t_2$ to $f_2 t_1$ | $0.887 \pm 0.023$ |
| $f_2 t_1$ to $f_1 t_2$ | $0.886 \pm 0.022$ |
| $f_2 t_2$ to $f_1 t_1$ | $0.877 \pm 0.014$ |

**Fig. 4** Multiplexed storage and quantum mode conversion for spatial encoded qutrit state using four temporal and spectral channels. **a** The multiplexed storage for spatial-qutrit state of $|\psi_1\rangle$ in the temporal and spectral DOF. **b** Fidelities for the qutrit states after storage. **c** The performance of the QMC with an encoded state of $|\psi_1\rangle$. **d** Fidelities for the qutrit state after mode conversion. The error bars for the fidelities correspond to one standard deviation caused by the statistical uncertainty of photon counts

in three DOF. The currently achieved multimode capacity is certainly not the fundamental limit for the physical system. $Pr^{3+}$ : $Y_2SiO_5$ has an inhomogeneous linewidth of 5 GHz, which can support more than 60 independent spectral modes. The number of temporal modes that can be achieved using the AFC protocol[17] is proportional to the number of absorption in the comb, which has already been improved to 50 in $Eu^{3+}$ :$Y_2SiO_5$[22]. There is no fundamental limit on the multimode capacity in the OAM DOF since it is independent on the AFC bandwidth. The capacity in this DOF is simply determined by the useful size of the memory in practice. We have recently demonstrated the faithful storage of 51 OAM spatial modes in a $Nd^{3+}$ :$YVO_4$ crystal[26]. The combination of these state-of-the-art technologies could result in a multimode capacity of $60 \times 50 \times 51 = 1,53,000$ modes. This large capacity could greatly enhance the data rate in memory-based quantum networks and in portable quantum hard drives with extremely long lifetimes[15].

The developed multiple-DOF quantum memory can serve as a QMC, which is a fundamental requirement for the construction of scalable networks based on multiplexed quantum repeaters. Although it is not demonstrated in the current work, mode conversion in the spatial domain should also be feasible using a high-speed digital micromirror device[38]. QMC can also find applications in linear optical quantum computations. One typical example is to solve the mode mismatch caused by fiber-loop length effects and the time jitter of the photon sources in a boson sampling protocol[39,40].

Quantum communication and quantum computation in a large-scale quantum network rely on the ability to faithfully store and manipulate photonic pulses carrying quantum information. The presented quantum memory can apply arbitrary temporal and spectral manipulations to photonic pulses in real time, which indicates that this single device can serve as a variable temporal

beam splitter[32,41] and a relative phase shifter[42] that enables arbitrary control of splitting ratio and phase for each output. Therefore, this device can perform arbitrary single-qubit operations[43]. Combining with the recent achievements on generation of heralded single photons[44,45], this device should provide the sufficient set of operations to allow for universal quantum computing in the Knill–Laflamme–Milburn scheme[46]. Our results are expected to find applications in large-scale memory-based quantum networks and advanced photonic information processing architectures.

## Methods

**AFC preparation.** We tailored the absorption spectrum of $Pr^{3+}$ ions to prepare the AFC using spectral hole burning[42]. The frequency of the pump light was first scanned over 16 MHz to create a wide transparent window in the $Pr^{3+}$ absorption line. Then, a 1.6 MHz sweep was performed outside the pit to prepare the atoms into the 1/2g state. The burn-back procedure created an absorbing feature of 2 MHz in width resonant with the 1/2g–3/2e transition, but simultaneously populated the 3/2g state, which, in principle, must be empty for spin-wave storage. Thus, a clean pulse was applied at the 3/2g–3/2e transition to empty this ground state. After the successful preparation of absorbing band in the 1/2g state, a stream of hole-burning pulses was applied on the 1/2g–3/2e transition. An AFC structure with a periodicity of $\Delta = 200$ kHz is prepared in this step. These pulses burned the desired spectral comb of ions on the 1/2g–3/2e transition and antiholes at the 3/2g–3/2e transition; thus, a short burst of clean pulses was applied to maintain the emptiness of the 3/2g state. For AFC preparation, the remaining 5/2g ground state is used as an auxiliary state, which stores those atoms which do not contribute to the AFC components. To reduce the noise generated by the control pulses during spin-wave storage, we applied 100 control pulses separated by 25 μs and another 50 control pulses with a separation of 100 μs after the preparation of the comb[21]. An example of the AFC with a periodicity $\Delta = 200$ kHz is illustrated in Fig. 3a. A detailed estimation of the structure and storage efficiency of the AFC memory is presented in Supplementary Note 1. The signal photons are mapped onto the AFC, leading to an AFC echo after a time $1/\Delta$. Spin-wave storage is achieved by applying two on-resonance control pulses to induce reversible transfer between the 3/2e state and 3/2g state before the AFC echo emission. The complete storage time is 12.68 μs in our experiment which includes an AFC storage time of 5 μs and a spin-wave storage time of 7.68 μs.

**a**

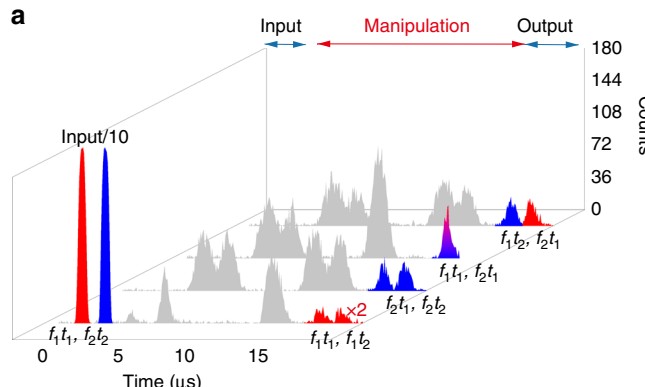

**b**

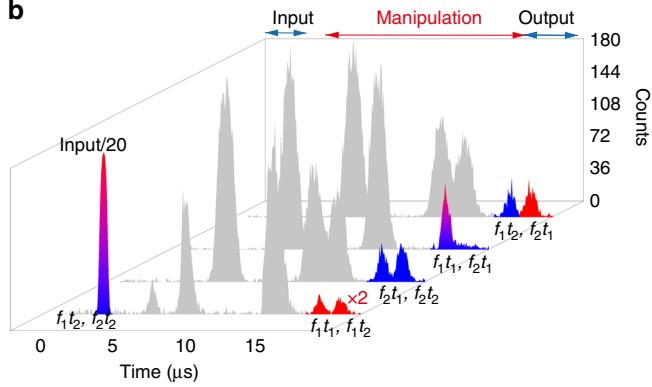

**Fig. 5** Arbitrary temporal and spectral manipulations in real time. Four typical operations are presented for two different input states. **a** The OAM qutrit state $|\psi_1\rangle$ is encoded on the $f_1t_1$ and $f_2t_2$ modes. The $f_1$ photons are marked as red color and the $f_2$ photons are marked as blue color. These operations, from up to down, correspond to a pulses sequencer, a multiplexer, a selective spectral shifter and a configurable beam splitter (the $f_2$ photons are filtered out), respectively. The little "×2" indicates that the integration time of temporal beam splitting is two times of the other operations. **b** The OAM qutrit state $|\psi_2\rangle$ is encoded on the $f_1t_2$ and $f_2t_2$ modes. These operations, from up to down, correspond to a demultiplexer, a pulses sequencer, a selective spectral shifter (the $f_1$ photons is frequency shifted to $f_2$ and retrieved at $t_1$, while the $f_2$ photons are retrieved at $t_2$) and a configurable beam splitter (the $f_2$ photons are filtered out), respectively. All of these operations can be determined after the photons have been absorbed into the quantum memory. The inputs are shifted earlier by 3 μs in the histograms for visual effects

### Table 1 Fidelities of qutrit states after temporal and spectral manipulations

| Input mode | Output mode | Fidelity |
|---|---|---|
| $\lvert\psi_1\rangle_{f_1t_1,\ f_2t_2}$ | $\lvert\psi_1\rangle_{f_1t_2,\ f_2t_1}$ | 0.881 ± 0.022 |
| | $\lvert\psi_1\rangle_{f_1t_1,\ f_2t_1}$ | 0.876 ± 0.022 |
| | $\lvert\psi_1\rangle_{f_2t_2,\ f_2t_2}$ | 0.897 ± 0.020 |
| | $\lvert\psi_1\rangle_{f_1t_1,\ f_1t_2}$ | 0.828 ± 0.019 |
| $\lvert\psi_2\rangle_{f_1t_2,\ f_2t_2}$ | $\lvert\psi_1\rangle_{f_1t_2,\ f_2t_1}$ | 0.896 ± 0.017 |
| | $\lvert\psi_2\rangle_{f_1t_1,\ f_2t_1}$ | 0.898 ± 0.013 |
| | $\lvert\psi_1\rangle_{f_2t_2,\ f_2t_2}$ | 0.898 ± 0.016 |
| | $\lvert\psi_1\rangle_{f_1t_1,\ f_1t_2}$ | 0.829 ± 0.025 |

The fidelity for the output mode of $\lvert\psi_1\rangle_{f_1t_1,\ f_1t_2}$ is a little lower than the others because of the less photon counts in each output caused by the temporal splitting operation. The error bars for the fidelities correspond to one standard deviation caused by the statistical uncertainty of photon counts.

**Filtering the noise.** In order to achieve a low noise floor, temporal, spectral, and spatial filter methods are employed. The input and control beams are sent to the MC in opposite directions with a small angular offset for spatial filtering. Temporal filtering is achieved by means of a temporal gate implemented with two AOM. This AOM gate temporally blocked the strong control pulses. This is important to avoid burning a spectral hole in the FC and to avoid blinding the single-photon detector. We used two 2-nm bandpass filters at 606 nm to filter out incoherent fluorescence noise. The spectral of the filter mode was achieved by narrow-band spectral filter in the FC (shown by the dashed black line in Fig. 3a), which is created by 0.8 MHz sweep around the input light frequency, leading to a transparent window of approximately 1.84 MHz due to the power broadening effect. Furthermore, the FC is implemented in a double-pass configuration to achieve high absorption.

**Quantum tomography.** To characterize the memory performance for three-dimensional OAM states, quantum process tomography for the quantum memory operation is performed. Reconstructing the process matrix $\chi$ of any three-dimensional state requires nine linearly independent measurements. We chose three OAM eigenstates and six OAM superposition states as our nine input states, which are listed as follows: $|L\rangle, |G\rangle, |R\rangle, (|L\rangle + |G\rangle)/\sqrt{2}, (|R\rangle + |G\rangle)/\sqrt{2}, (i|L\rangle + |G\rangle)/\sqrt{2}, (-i|R\rangle + |G\rangle)/\sqrt{2}, (|L\rangle + |R\rangle)/\sqrt{2}$, and $(|L\rangle - i|R\rangle)/\sqrt{2}$[26]. The complete operators for the reconstruction of the matrix $\chi$ are as follows:[26,47]

$$\lambda_1 = \begin{bmatrix} 1 & 0 & 0 \\ 0 & 1 & 0 \\ 0 & 0 & 1 \end{bmatrix} \tag{1}$$

$$\lambda_2 = \begin{bmatrix} 0 & 1 & 0 \\ 1 & 0 & 0 \\ 0 & 0 & 0 \end{bmatrix} \tag{2}$$

$$\lambda_3 = \begin{bmatrix} 0 & -i & 0 \\ i & 0 & 0 \\ 0 & 0 & 0 \end{bmatrix} \tag{3}$$

$$\lambda_4 = \begin{bmatrix} 1 & 0 & 0 \\ 0 & -1 & 0 \\ 0 & 0 & 0 \end{bmatrix} \tag{4}$$

$$\lambda_5 = \begin{bmatrix} 0 & 0 & 1 \\ 0 & 0 & 0 \\ 1 & 0 & 0 \end{bmatrix} \tag{5}$$

$$\lambda_6 = \begin{bmatrix} 0 & 0 & -i \\ 0 & 0 & 0 \\ i & 0 & 0 \end{bmatrix} \tag{6}$$

$$\lambda_7 = \begin{bmatrix} 0 & 0 & 0 \\ 0 & 0 & 1 \\ 0 & 1 & 0 \end{bmatrix} \tag{7}$$

$$\lambda_8 = \begin{bmatrix} 0 & 0 & 0 \\ 0 & 0 & -i \\ 0 & i & 0 \end{bmatrix} \tag{8}$$

$$\lambda_9 = \frac{1}{\sqrt{3}}\begin{bmatrix} 1 & 0 & 0 \\ 0 & 1 & 0 \\ 0 & 0 & -2 \end{bmatrix} \tag{9}$$

Here, $\lambda_1$ is the identity operation. The process matrix $\chi$ can be expressed on the basis of $\lambda_i$ and maps an input matrix $\rho_{in}$ onto the output matrix $\rho_{out}$[47,48]. Density matrices of the input states $\rho_{in}$ and density matrices of the output states $\rho_{out}$ are reconstructed using quantum state tomography[47,48]. For a given qutrit state $(|\psi\rangle)$, we used $\rho = |\psi\rangle\langle\psi|$ to reconstruct the density matrix $\rho$ from the measurement results. As examples, we presented graphical representations of the reconstructed density matrices for the OAM qutrit states $|\psi_1\rangle$ and $|\psi_2\rangle$ in Supplementary Note 3. The fidelities of the output states with respect to the input states can be calculated based on the reconstructed density matrices using the formula

$$Tr\left(\sqrt{\sqrt{\rho_{out}}\rho_{in}\sqrt{\rho_{out}}}\right)^2.$$

**Data availability**. The data that support the findings of this study are available from the corresponding authors on request.

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

## Acknowledgments

This work was supported by the National Key R&D Program of China (No. 2017YFA0304100), the National Natural Science Foundation of China (Nos. 61327901, 11774331, 11774335, 61490711, 11504362, and 11654002), Anhui Initiative in Quantum Information Technologies (No. AHY020100), Key Research Program of Frontier Sciences, CAS (No. QYZDY-SSW-SLH003), the Fundamental Research Funds for the Central Universities (Nos. WK2470000023 and WK2470000026).

## Author contributions

Z.Q.Z. and C.F.L. designed experiment. T.S.Y. and Z.Q.Z. carried out the experiment assisted by Y.L.H., X.L., Z.F.L., P.Y.L., Y.M., C.L., P.J.L., Y.X.X., J.H., and X.L. T.S.Y. and Z.Q.Z. wrote the paper with input from other authors. C.F.L. and G.C.G supervised the project. All authors discussed the experimental procedures and results.

## Additional information

**Competing interests:** The authors declare no competing interests.

