## [Peer Review File · Nature Communications]

Reviewers' comments:

Reviewer #1 (Remarks to the Author):

The manuscript presents a series of experiments of storage of qutrit and qubit states encoded in coherent states of light into and out of a rare-earth ion doped crystal memory using the spin-echo AFC quantum memory protocol. This demonstration has an important application in the realization of long-distance quantum communication based on quantum repeaters and in linear optics quantum computing. Elements of the work have been demonstrated previously, but the manuscript significantly expands the capability for multimode storage in several degrees of freedom and interestingly involves the spatial degree of freedom. The manuscript is overall well written and the results and experimental details are clearly presented. A major caveat is the fact that the results not performed with true single photons for encoding the quantum information and the statistics of the coherent state are not accounted for in the analysis that the authors perform (see 'Major Comments' below). Based on this, I am somewhat in favour of publishing the manuscript in Nature Communications, if the authors can address the comments listed below.

Major Comments:

Abstract and main text: The experiments were performed with attenuated laser pulses with a mean photon number around 1 per pulse. First of all, this fact must be clearly disclosed in the Abstract. The tricky question is whether storage of attenuated laser pulses constitutes a proof of that a memory operates in the quantum regime. Various methods to thoroughly verify the quantumness of a memory using attenuated laser pulses are outlined in e.g. [Gundogan et al., PRL 108, 190504] and [Sinclair et al., PRL 113, 053603]. This manuscript does not employ any of these methods. Although it could be argued that previous experiments have thoroughly established that the AFC protocol does preserve the stored quantum states, it would be very useful to evaluate the fidelity of e.g. the process matrix according to the criterion in [PRL 108, 190504].

Main text, page 3, 2nd paragraph, line 19: The minimal signal to noise ratio is argued to be 11.90, however it is not clear how this is calculated. There can be different ways to define an SNR. In the simplest case one takes the counts in the diagonal term and then locates the largest corresponding peak over the range of input modes. This then says, given a particular output mode is detected after the memory, what is the chance of the detection being caused by the wrong input state. However, the SNR should be seen in light of the degrees-of-freedom used for multiplexing and those used for encoding the qubit/qutrit state. In other words, if the temporal and spectral modes are used for multiplexing, then in an application one would simultaneously store qubit/qutrit states in all degrees of freedom. This notion is the basis of the mode conversion presented in Fig. 3c. Hence, for the SNR one should estimate the maximal error counts in a particular qubit/qutrit state detection over all possible combinations of encoding other qubit/qutrit states simultaneously in the other multiplexed modes. I would say this is the most relevant approach to estimate the SNR and in any case the authors should specify how it is calculated.

Minor comments:

Abstract:

lines 6-7: The rationale behind the statement "[...], highly multi- mode quantum memories will be required" should be elaborated. It is not clear from a non-specialist reader that this refers to the memory's capability to be incorporated in a multiplexed repeaters scheme.

lines 14-15: It is not clear to me what is meant by the term 'scalability' in the sentence "[...] we

create a multiple- degree-of-freedom quantum memory with high scalability”?

Main text:

1st paragraph, line 2: Add ‘long-distance’ and ‘optical quantum’ to the sentence “such as long-distance quantum key distribution [1] and optical quantum computing”

page 1, 1st paragraph, line 6: The term “on the ground” is vague. Authors might as well use a term such as “[...] via ground based optical fibers [...]”

page 1, 2nd paragraph, line 1: Cut the plural ‘s’ from “degree-of-freedom”.

page 1, 3rd paragraph, line 10: Remove the word ‘recently’ (it appears twice in the sentence)

page 3, 1st paragraph, line 7: The symbols for κ_{input} and κ_{output} are listed in the wrong order. It should be “[...] between χ_{output} and κ_{input} (κ_{ideal}) is [...]”.

page 3, 2nd paragraph, line 7: Make a reference to Fig. 3a i.e. “[...] interval of 80 MHz between them (see Fig. 3a) to achieve [...]”

page 3, 2nd paragraph, line 12: The authors say that two temporal modes are employed. It would be helpful to specify if this is done by changing the spacing of the control pulses.

page 3, 2nd paragraph, line 12: The authors mention that only two temporal modes are used so as to “minimize noise”. What sort of noise i.e. what is the origin (control pulses?) and how do the two pulses minimize it?

page 3, 2nd paragraph, line 14: Remove the words “well-defined” (‘defined’ appears later in the sentence) and ‘of’.

page 3, Fig. 2a: It would be useful to label the two black detection pulses - I guess they are caused the control pulses.

Reviewer #2 (Remarks to the Author):

The manuscript presents the storage of qutrits encoded in orbital angular momentum (OAM) of light with a classical multiplexing in time and frequency (2 x 2 modes). The atomic medium is a rare earth doped crystal (Pr:YSO), and they use the well known atomic frequency comb (AFC) protocol with the spin-wave storage step (meaning on-demand retrieval), with which achieving a reasonable signal to noise ratio is a bit more challenging. The storage performances seem to be in the average, but the authors highlights the multiplexing aspect, and they use more than one degree of freedom (DOF) to do so. There is nevertheless a lack of clarity in distinguishing classical DOF and quantum DOF in this work, which is problematic (see point 1 below). The presentation of the quantumness of the memory (preservation of the OAM qutrit) is not clearly presented. When they give fidelities (state or process), they do not compare with the classical limit, which is supposed to be the only way to show the quantumness in this regime. They also demonstrate manipulations of the classical DOF (time and frequency mode operations), which preserve the quantum state. Below I list a number of more detailed remarks.

One of the main problem is the presentation of quantum memory (QM) for different DOF. To me,

we have to be careful with this claim. The authors compare their work with storage of hyper-entanglement, for example the work of Tiranov et al. (Optica 2015), where quantum superposition are observed in two DOF, and analyzed for each DOF, varying the basis of analysis in the other DOF (and the results shouldn't depend on the chosen basis). The manuscript gives the impression that the authors mix DOF of entanglement and DOF for classical multiplexing. Their QM is quantum only for one DOF (OAM), and there is nothing quantum for the frequency and time encoding DOF.

What is a quantum mode converter? They should define it, since it is not defined in ref 30, the review on QM by Heshami et al. (J. Mod. Opt. 63, 2005-2028 (2016)).

"and thus can serve as a real-time sequencer [13], a real-time multiplexer/demultiplexer [31], a real-time beam splitter [32], a random-access memory [33], a real-time frequency shifter [34], a real-time temporal/spectral filter [31], among other functionalities." The authors don't explain these functions, (in particular random-access memory does not seem trivial) and never come back to those, after the presentation of the results.

The authors should give the efficiency of the QM (efficiency of the storage and of the coupling, filtering step, ...)

The signal to noise ratio (SNR) is quite high (~40). From what I remember, the SNR for spin-wave storage are usually around 1.

Page 3, first column: "Moreover, we note that the memory performance for superposition states of $|L\rangle$ and $|R\rangle$ is much better than that achieved here (as detailed in Supplementary Section II)." I don't understand what they mean. The authors should rephrase this.

The minimal SNR is 3 times lower when they perform the multimode storage. Why is it lower? Does the factor 3 correspond to the 3 modes? Does it scale like that? (10 modes = SNR/10) How to remedy this problem?

Figure 2: What is the basis of the density matrix representing the storage process? Shouldn't it be 4x4?

General comment: All the fidelities must be compared to (and should be above) the classical limit for their mean photon number. Otherwise, they cannot claim that the memory is quantum.

Again, page 3, first column: " $2 \times 2 \times 3 = 12$ modes in total" But in this work, the multiplexing is only in frequency and time, so $2 \times 2 = 4$ channels. Moreover, they should show measurement of the average fidelity of the OAM qutrit for each classical "channel".

About the "Arbitrary manipulations in real time", part (page 4, first column), I don't really get the interest in encoding the same state in 2 modes and shifting the frequency or time mode of them, probably because I'm not an expert on quantum computing. The authors should cite a paper where those operations are mentioned. Furthermore, why not making operations on the 2 different OAM states ($|\phi_1\rangle$ and $|\phi_2\rangle$)?

The presentation of the results of the manipulation in Figure 3 is really hard to read.

To conclude, the work is interesting but not presented clearly, and some imprecisions are made. In consequence, I can't recommend the publication in Nature Communications.

Reviewer #3 (Remarks to the Author):

In the paper 'Multiplexed storage and real-time manipulation based on a multiple-degree-of-freedom quantum memory', the authors present a scheme in which photonic quantum information encoded in three different degrees of freedom can be stored and manipulated in a quantum memory.

The three different degrees of freedom that are used are the orbital angular momentum (OAM), the time and the frequency. The quantum memory protocol is the spin-wave atomic frequency comb protocol, which the authors use to store coherent states at the single photon level with high signal-to-noise ratio.

I consider that the developments that are proposed in this article are very timely, as densification of quantum information encoding and manipulation is a key resource for the development of large-scale quantum networks.

However, two major points should in my opinion be addressed before I can make my final decision about the possibility to publish this work in Nature Communications.

- My first concern comes from the estimation of the fidelities between the input and manipulated output states, in the case of conversion or arbitrary manipulation. Indeed, all the fidelities that are presented in tables I and II in the article only concern the fidelity in the degree of freedom that is not manipulated (the OAM), which in my opinion is not relevant for characterizing the transformation. Other tools like process matrices in the time-frequency space (dimension 4 here) would be more adapted to prove that the transformation that is performed is indeed the one that is expected.

- The second important point is related to the fidelity of the process matrices that the authors present for the OAM. It is very surprising to me to calculate the fidelity between two process matrices in this context (χ_{input} and χ_{output} , or χ_{ideal} and χ_{output}), given that we want to characterize the process associated to the quantum memory only. Indeed, if I understood correctly, χ_{output} is the process matrix calculated with the output density matrices of the whole process (preparation of the OAM and memory) and the input ideal density matrices (pure states), whereas it should be calculated using the density matrices in input of the quantum memory (mixed states, taking into account the imperfect preparation). The relevant number would then be the 'identity component' of this process matrix.

Instead of this, the authors compare how similar are the process matrices with and without the quantum memory by calculating a fidelity between them. This fidelity, in my opinion, is not equivalent and less relevant than the previously mentioned 'identity component'.

In addition to these two major issues, a few points also raised questions during my reviewing process:

- Following the second point mentioned above, two numbers are in my opinion missing in the text: what are the fidelities of the prepared OAM qutrit states with the ideal ψ_1 and ψ_2 states? And what is limiting in this case? For instance, on figure S4 it looks like the $|G\rangle\langle G|$ component is higher than all the others: is it due to preparation imperfection or to the action of the memory?

- On figure 2b: could the authors precise what the λ_i operators are (even in the suppl. mat.), and give an intuition why the imperfections seem to mainly come from λ_4 and λ_5 ?

- Could the authors explain why they chose a spin duration of 7.68 μs ? Has this particular number been chosen for noise issues?

- The combs that are presented in the paper allow to reach efficiencies which are 0.5% lower than in the case of a unique comb (section I of suppl. mat.): is there a particular reason for this small decrease in the efficiency? Is the inhomogeneous profile responsible for this small drop (lower optical depth in order to match both efficiencies)? Also, regarding the combs: the authors use two temporal modes in combs that possess ten teeth. This means that overall, approximately ten

modes could be used. Given that the SNR is high (almost 40 in their case), why did the authors limit themselves to two modes, and how dramatic is the decrease of the SNR if this number of modes is increased?

- The authors claim that two fidelities in table 2 are lower, due to 'less photon counts in each output'. But a decrease in the photon counts should decrease the precision and not the fidelity: here the fidelity is decreased, well below the error bars that are presented. Could the authors comment on this?

- A 'classical benchmark' is mentioned regarding the states manipulation, before the final discussion. The authors claim to be well above it: could they precise which limit is mentioned here? More precisely, as the states that are used are weak coherent states, the limit fidelity of $2/3$ should strictly speaking not be used and a more complete criterion must be chosen.

Eventually, I also found some typos in the article:

- In the discussion about fidelities between process matrices, (χ_{ideal}) should be next to χ_{input} and not χ_{output} , and 'resp.' could be used to clarify the two fidelities that are presented.

- In the 'Arbitrary manipulation in real time', I think that the 'exchange of the readout times for the f_1 and f_2 photons' should read $f_1 t_2, f_2 t_1$ both in the text and in table II, as correctly written on figure 4.

- The z axes in figure S4 are misleading, as χ is usually used for process matrices, and ρ_{out} is the name that is mentioned in the methods.

- First paragraph, line 12: 'use' should be 'uses'

- Fourth paragraph, line 7: 'In addition to the increasing' should be 'In addition to increasing'

- Methods, line 3: 'Refs.' should be 'Ref.'

- [27]: 'Parigi1' should be 'Parigi'

- Caption of figure S4: a parenthesis is missing for state ψ_1 .

Response to Referee #1 -- NCOMMS-17-29988

We thank the Referee for making useful suggestions and comments and reply below, in blue font, to the posed criticisms.

The manuscript presents a series of experiments of storage of qutrit and qubit states encoded in coherent states of light into and out of a rare-earth ion doped crystal memory using the spin-echo AFC quantum memory protocol. This demonstration has an important application in the realization of long-distance quantum communication based on quantum repeaters and in linear optics quantum computing. Elements of the work have been demonstrated previously, but the manuscript significantly expands the capability for multimode storage in several degrees of freedom and interestingly involves the spatial degree of freedom. The manuscript is overall well written and the results and experimental details are clearly presented. A major caveat is the fact that the results not performed with true single photons for encoding the quantum information and the statistics of the coherent state are not accounted for in the analysis that the authors perform (see 'Major Comments' below). Based on this, I am somewhat in favor of publishing the manuscript in Nature Communications, if the authors can address the comments listed below.

We thank the Referee for these very positive comments.

Abstract and main text: The experiments were performed with attenuated laser pulses with a mean photon number around 1 per pulse. First of all, this fact must be clearly disclosed in the Abstract.

We thank the Referee for the helpful suggestions. We have pointed out that our experiments were performed with weak coherent pulses of 1 photon per pulse in the revised Abstract.

The tricky question is whether storage of attenuated laser pulses constitutes a proof of that a memory operates in the quantum regime. Various methods to thoroughly verify the quantumness of a memory using attenuated laser pulses are outlined in e.g. [Gundogan et al., PRL 108, 190504] and [Sinclair et al., PRL 113, 053603]. This manuscript does not employ any of these methods. Although it could be argued that previous experiments have thoroughly established that the AFC protocol does preserve the stored quantum states, it would be very useful to evaluate the fidelity of e.g. the process matrix according to the criterion in [PRL 108, 190504].

We thank the Referee for the helpful suggestions. According to Referee's suggestions, we have employed the abovementioned criterion to infer the

quantumness of our memory [PRL 108, 190504 (2012)]. We have presented the results in Supplementary Information section IV. Using this criterion, the achieved fidelity is significantly greater than the limit for a classical memory. These results demonstrate the quantum nature of our memory.

Main text, page 3, 2nd paragraph, line 19: The minimal signal to noise ratio is argued to be 11.90, however it is not clear how this is calculated. There can be different ways to define an SNR. In the simplest case one takes the counts in the diagonal term and then locates the largest corresponding peak over the range of input modes. This then says, given a particular output mode is detected after the memory, what is the chance of the detection being caused by the wrong input state. However, the SNR should be seen in light of the degrees-of-freedom used for multiplexing and those used for encoding the qubit/qutrit state. In other words, if the temporal and spectral modes are used for multiplexing, then in an application one would simultaneously store qubit/qutrit states in all degrees of freedom. This notion is the basis of the mode conversion presented in Fig. 3c. Hence, for the SNR one should estimate the maximal error counts in a particular qubit/qutrit state detection over all possible combinations of encoding other qubit/qutrit states simultaneously in the other multiplexed modes. I would say this is the most relevant approach to estimate the SNR and in any case the authors should specify how it is calculated.

Thanks for your suggestions. We need to elaborate that the SNR of 11.90 in our previous manuscript is calculated as one takes the counts in the diagonal term as the signal and then locates the large peaks over the range of **output** modes as the noise. This benchmarks the detection error caused by the wrong output modes. But as the Referee says, one can also take the counts in the diagonal term as the signal and then locates the large peaks over the range of **input** modes as the noise. This benchmarks the detection error caused by the wrong input modes. These results together indicate the crosstalk between these different modes. We clarified the definition on SNR in the 2nd paragraph on page 3 of the revised manuscript.

We agree with the Referee that the best approach to estimate the SNR is that one estimates the maximal error counts in a particular qutrit detection over all possible combinations of encoding other qutrit states simultaneously in the other multiplexed modes. However, limited by the slow response time of the spatial light modulator (~10ms), it is not possible to generate many different qutrit states for all the input modes. Therefore, we cannot estimate the SNR using this approach. Nevertheless, we can estimate the maximal error counts in a particular output mode caused by the other input modes from the results presented Fig. 4a. The SNR estimated from the mode crosstalk is approximately 15.2.

Minor comments:

Abstract:

lines 6-7: The rationale behind the statement “[...], highly multimode quantum memories will be required” should be elaborated. It is not clear from a non-specialist reader that this refers to the memory’s capability to be incorporated in a multiplexed repeaters scheme.

Following the Referee’s suggestions, we have revised the Abstract. We use “The faithful storage and coherent manipulation of quantum states with matter-systems enable the construction of large-scale quantum networks based on quantum repeater. To achieve useful communication rates, highly multimode quantum memories will be required to construct a multiplexed quantum repeater.” to elaborate the statement. We hope this will be clear for a non-specialist reader.

lines 14-15: It is not clear to me what is meant by the term ‘scalability’ in the sentence “[...] we create a multiple- degree-of-freedom quantum memory with high scalability”?

By using ‘scalability’, we wish to claim that the quantum memory with multiplexing in multiple DOF has a large multimode capacity. To avoid any misunderstandings, we change the word to ‘multimode capacity’ in the revised Abstract. We elaborated the discussion on multimode capacity in the first paragraph in the Discussion section.

Main text:

1st paragraph, line 2: Add ‘long-distance’ and ‘optical quantum’ to the sentence “such as long-distance quantum key distribution [1] and optical quantum computing”

Thanks. We have added these phrases in the revised manuscript based on your suggestions.

page 1, 1st paragraph, line 6: The term “on the ground” is vague. Authors might as well use a term such as “[...] via ground based optical fibers [...]”

Thanks. This explanation was added in the revised manuscript based on your suggestions.

page 1, 2nd paragraph, line 1: Cut the plural ‘s’ from “degree-of-freedom”.

Corrected.

page 1, 3rd paragraph, line 10: Remove the word 'recently' (it appears twice in the sentence)

Corrected.

page 3, 1st paragraph, line 7: The symbols for χ_{input} and χ_{output} are listed in the wrong order. It should be "[...] between χ_{output} and χ_{input} (χ_{ideal}) is [...]"

Corrected.

page 3, 2nd paragraph, line 7: Make a reference to Fig. 3a i.e. "[...] interval of 80 MHz between them (see Fig. 3a) to achieve [...]"

We thank the Referee for pointing out to us this. We have added this reference in the revised manuscript.

page 3, 2nd paragraph, line 12: The authors say that two temporal modes are employed. It would be helpful to specify if this is done by changing the spacing of the control pulses.

In this case, we did not change the spacing of the control pulses to realize two temporal modes. We combined two beams of control pulses and two beams of input pulses using two beam splitters in front of the cryostat. Therefore, two temporal modes are totally independent.

page 3, 2nd paragraph, line 12: The authors mention that only two temporal modes are used so as to "minimize noise". What sort of noise i.e. what is the origin (control pulses?) and how do the two pulses minimize it?

The noise is caused by the control pulses. Increasing the number of modes, the time interval between the last control pulse and the first output signal pulse will be reduced. This will lead to increased noise and we limit our experiment to two modes. However, this limit can be overcome by long AFC echo times [Phys. Rev. A 93,032327 (2016)]. This explanation was included in the 2nd paragraph on page 3 of the revised manuscript.

page 3, 2nd paragraph, line 14: Remove the words "well-defined" ('defined' appears later in the sentence) and 'of'.

Corrected.

page 3, Fig. 2a: It would be useful to label the two black detection pulses - I guess they are caused the control pulses.

Corrected.

We sincerely thank the Referee for reading our manuscript carefully. We have carefully checked the manuscript and corrected all the typos accordingly.

Response to Referee #2 -- NCOMMS-17-29988

We thank the Referee for reading our manuscript carefully and for making useful suggestions and comments. We think that there are some issues that we have not explained clearly enough in the original version of our paper. Now we present new experimental results and reply to the referee's criticisms below using blue font.

The manuscript presents the storage of qutrits encoded in orbital angular momentum (OAM) of light with a classical multiplexing in time and frequency (2 x 2 modes). The atomic medium is a rare earth doped crystal (Pr:YSO), and they use the well-known atomic frequency comb (AFC) protocol with the spin-wave storage step (meaning on-demand retrieval), with which achieving a reasonable signal to noise ratio is a bit more challenging. The storage performances seem to be in the average, but the authors high lights the multiplexing aspect, and they use more than one degree of freedom (DOF) to do so. There is nevertheless a lack of clarity in distinguishing classical DOF and quantum DOF in this work, which is problematic (see point 1 below). The presentation of the quantumness of the memory (preservation of the OAM qutrit) is not clearly presented. When they give fidelities (state or process), they do not compare with the classical limit, which is supposed to be the only way to show the quantumness in this regime. They also demonstrate manipulations of the classical DOF (time and frequency mode operations), which preserve the quantum state. Below I list a number of more detailed remarks.

We thank the Referee's for the detailed evaluations and suggestions. The fidelity issue is discussed in the reply below.

One of the main problem is the presentation of quantum memory (QM) for different DOF. To me, we have to be careful with this claim. The authors compare their work with storage of hyper-entanglement, for example the work of Tiranov et al. (Optica 2015), where quantum superposition are observed in two DOF, and analyzed for each DOF, varying the basis of analysis in the other DOF (and the results shouldn't depend on the chosen basis). The manuscript gives the impression that the authors mix DOF of entanglement and DOF for classical multiplexing. Their QM is quantum only for one DOF (OAM), and there is nothing quantum for the frequency and time encoding DOF.

We thank the Referee for the helpful suggestions. We have strictly distinguished the classical “DOF” and quantum “DOF” throughout the revised manuscript.

To demonstrate the ability of multiplexed storage over 3 classical DOF, we have supplemented a new experiment, as shown in Fig. 3b and Fig. 3c of the revised manuscript. The spatial multiplexing is now realized by using three independent paths, which are denoted as s_1 , s_2 , s_3 spatial modes. It's true that we only demonstrate that the OAM DOF can carry quantum information in the current work although the other two DOF can also be employed as quantum DOF in principle. We pointed out in the 3rd paragraph on page 3: “Here, the temporal, spectral and spatial DOF are employed as classical DOF for multiplexing. One can choose any DOF to carry quantum information. As a typical example, now we use the temporal and spectral DOF for multiplexing and each channel is encoded with spatial qutrit state...”

What is a quantum mode converter? They should define it, since it is not defined in ref 30, the review on QM by Heshami et al. (J. Mod. Opt. 63, 2005-2028 (2016)).

The quantum mode converter (QMC) can transfer photonic pulses to a target temporal or spectral mode without distorting the photonic quantum states. Here the quantum states are encoded in the orbital angular momentum (OAM) space. This device enables mode conversion between any classical modes while preserving the OAM superposition states for applications in high-dimensional quantum information processing protocols. Careful optical mode-matching is essential for quantum information transfer between systems [J. Mod. Opt. 63, 2005-2028 (2016)]. As discussed in the manuscript, this device can ensure that the photons participating in a joint measurement, after being retrieved from any quantum memory, are indistinguishable, as is required for, e.g., a Bell-state measurement [Phys. Rev. Lett. 113, 053603 (2014)]. QMC can also find applications in linear optical quantum computations. One typical example is to solve the mode mismatch caused by fiber-loop length effects and the time jitter of the photon sources in a boson sampling protocol [Phys. Rev. Lett. 113, 120501(2014), Phys. Rev. A. 92, 052319 (2015)]. We included the definition and discussion on QMC in the last paragraph on page 3.

“and thus can serve as a real-time sequencer [13], a real-time multiplexer/demultiplexer [31], a real-time beam splitter [32], a random-access memory [33], a real-time frequency shifter [34], a real-time temporal/spectral filter [31], among other functionalities.” The authors don't explain these functions, (in particular random-access memory does not seem trivial) and never come back to those, after the presentation of the results.

Thanks for your suggestions. We have explained and displayed these functions in the caption of Fig. 5 in the revised manuscript.

Input pulses occupying different spectral modes and different temporal modes (input: f_1t_1, f_2t_2) are mapped onto the processor as shown in Fig. 5a.

Their readout times can be exchanged (output: f_1t_2, f_2t_1). Here the processor serves as a real-time sequencer. The real-time sequencer can store and recall the pulses in any order. It can create a random access memory for time-bin encoded quantum information [Nature 461, 241–245 (2009)]. The random-access memory is not directly demonstrated using time-bin qubits in the current work; therefore we deleted the random-access memory in the text.

The f_1 and f_2 photons can be readout at the same time t_1 (output: f_1t_1, f_2t_1). Here the processor serves as a real-time multiplexer.

The frequency of f_1 photons is shifted to f_2 but the frequency of f_2 photons is unchanged (output: f_2t_1, f_2t_2). Here the processor serves as a real-time frequency shifter.

The f_1 photons are divided into two temporal modes but the f_2 photons are filtered out (output: f_1t_1, f_1t_2). Here the processor serves as a real-time beam splitter and a real-time spectral filter.

The authors should give the efficiency of the QM (efficiency of the storage and of the coupling, filtering step, ...)

We thank the Referee for the helpful suggestion. We have presented these efficiencies of the setup in Supplementary Information section I.

The signal to noise ratio (SNR) is quite high (~ 40). From what I remember, the SNR for spin-wave storage are usually around 1.

Due to the strong noise generated by the control pulses, the SNR of spin wave storage in solids is typically low. Filter crystal and Fabry-Perot filter are employed to filter out unwanted noises [Phys. Rev. Lett. 114, 230501 (2015), Phys. Rev. Lett. 114, 230502 (2015)]. SNR of approximately 10 were achieved for quantum storage at the single photon level. We achieved higher SNR by using a double-passed filter crystal and two narrow band pass filters. The filter crystal is employed to filter noise in resonance with the absorption of the filter crystal and the narrow band pass filter is employed to filter the noise originated from fluorescence to other energy levels.

Page 3, first column: “Moreover, we note that the memory performance for superposition states of $|L\rangle$ and $|R\rangle$ is much better than that achieved here (as detailed in Supplementary Section II).” I don’t understand what they mean. The authors should rephrase this.

We measured the visibility of superposition states of $|L\rangle$ and $|R\rangle$, which is slightly higher than the fidelity of the memory process for all three dimensions. This is because that the storage efficiency is balanced for the symmetrical LG modes but is not balanced for all three considered spatial modes. The efficiencies for the Gaussian and LG modes are not balanced due to the Gaussian mode has a smaller diameter than that of the LG modes. Moreover, further increasing the beam diameter or using a super-Gaussian spatial profile for the pump/control light could improve the memory performance for high dimensional states as we already demonstrated in two-level AFC storage [Phys. Rev. Lett. 115, 070502 (2015)]. We now rephrase these sentences so that the readers can understand this point easily.

The minimal SNR is 3 times lower when they perform the multimode storage. Why is it lower? Does the factor 3 correspond to the 3 modes? Does it scale like that? (10 modes = SNR/10) How to remedy this problem?

We are sorry for the misleading words in the original manuscript. At first, we measured the SNR of ~40 for the spin wave storage of the Gaussian mode ($|G\rangle$). Here, the SNR is calculated by taking the output with input as the signal and the output without input as the noise.

While for the multimode storage, the SNR is estimated from the mode crosstalk (~11). The mode crosstalk is calculated by taking the diagonal counts as the signal and the largest counts in other output modes as the noise. This is different from the SNR in single-mode storage. The mode crosstalk shows little dependence on the number of modes.

We have clarified the definition on SNR in the 2nd paragraph on page 3 of the revised manuscript.

Figure 2: What is the basis of the density matrix representing the storage process? Shouldn't it be 4x4?

To completely characterize the memory performance in three dimensions of the OAM DOF, we performed the quantum process tomography for qutrit operations. These operators are the complete operators in three dimensional Hilbert space [Phys. Rev. A 66, 012303 (2002), Phys. Rev. Lett. 115 ,070502 (2015)]. Therefore, it should be 9x9. The operators are given as follows:

$$\lambda_1 = \begin{bmatrix} 1 & 0 & 0 \\ 0 & 1 & 0 \\ 0 & 0 & 1 \end{bmatrix}; \lambda_2 = \begin{bmatrix} 0 & 1 & 0 \\ 1 & 0 & 0 \\ 0 & 0 & 0 \end{bmatrix}; \lambda_3 = \begin{bmatrix} 0 & i & 0 \\ -i & 0 & 0 \\ 0 & 0 & 0 \end{bmatrix}; \lambda_4 = \begin{bmatrix} 1 & 0 & 0 \\ 0 & -1 & 0 \\ 0 & 0 & 0 \end{bmatrix}; \lambda_5 = \begin{bmatrix} 0 & 0 & 1 \\ 0 & 0 & 0 \\ 1 & 0 & 0 \end{bmatrix}; \lambda_6 = \begin{bmatrix} 0 & 0 & i \\ 0 & 0 & 0 \\ i & 0 & 0 \end{bmatrix}; \lambda_7 = \begin{bmatrix} 0 & 0 & 0 \\ 0 & 0 & 1 \\ 0 & 1 & 0 \end{bmatrix}; \lambda_8 = \begin{bmatrix} 0 & 0 & 0 \\ 0 & 0 & -i \\ 0 & i & 0 \end{bmatrix}; \lambda_9 =$$

$\frac{1}{\sqrt{3}} \begin{bmatrix} 1 & 0 & 0 \\ 0 & 1 & 0 \\ 0 & 0 & -2 \end{bmatrix}$. We have presented these operators in Methods in the revised manuscript.

General comment: All the fidelities must be compared to (and should be above) the classical limit for their mean photon number. Otherwise, they cannot claim that the memory is quantum.

We thank the Referee for the helpful suggestions. To demonstrate the quantum behavior of our memory, the measured fidelity is compared with the highest fidelity achievable with a measure-and-prepare approach, taking into account the Poissonian statistics of the input states and the finite memory efficiency [Nature 489, 541 (2012), Nature Photonics 8, 234 (2014), PRL 108, 190504 (2015)]. We have presented these results in Supplementary Information section IV. These results demonstrate the quantum nature of our memory.

Again, page 3, first column: “2 2 3 = 12 modes in total” But in this work, the multiplexing is only in frequency and time, so 2 x 2 = 4 channels. Moreover, they should show measurement of the average fidelity of the OAM qutrit for each classical “channel”.

We thank the Referee for the helpful suggestions. To demonstrate the ability of multiplexed storage over 3 classical DOF, three independent spatial modes are involved in the revised experiment. The experimental results are presented in Fig. 3b and Fig. 3c.

According to Referee’s suggestion, we have demonstrated the multiplexed storage for qutrit state in the temporal and spectral DOF. We have measured the memory fidelity for spatial qutrit states in four temporal and spectral “channels”. The experimental results are shown in the Fig. 4a and Fig. 4b.

About the “Arbitrary manipulations in real time”, part (page 4, first column), I don’t really get the interest in encoding the same state in 2 modes and shifting the frequency or time mode of them, probably because I’m not an expert on quantum computing. The authors should cite a paper where those operations are mentioned. Furthermore, why not making operations on the 2 different OAM states (ϕ_1 and ϕ_2)?

Temporal shifter can be used as a sequencer [Nature 461, 241–245 (2009)]. Frequency shifter allows render photons indistinguishable without the need for a variable storage time in construction of a multiplexed quantum repeater [Phys. Rev. Lett. 113, 053603 (2014)]. The frequency-shifter could act as a universal adapter for accessing and distributing the quantum states of different quantum systems.

Limited by the slow response time of the spatial light modulator (~10ms), it is not possible to generate different qutrit states for the two input modes. Therefore, we didn't make operation on the two different OAM states in a single experiment. Although it is not demonstrated in the current work, operations on different OAM states and arbitrary manipulation in the spatial domain should also be feasible using a high-speed digital micro-mirror device [Opt. Express 20, 29269-29282 (2016)].

The presentation of the results of the manipulation in Figure 3 is really hard to read.

To make the information in Fig. 3 easily accessible for readers, we divided Fig. 3 into Fig. 3 and Fig. 4 and gave more detailed explanations in the revised manuscript. Fig. 3a shows the double AFC structure (red) in the memory crystal and the filter structure (black) in the filter crystal. Fig. 3b presents the input part of the multiplexed memory. s_1 , s_2 and s_3 carry different spatial information and correspond to the spatial modes s_1 , s_2 and s_3 in Fig. 3c. Fig. 3c shows multiplexed storage using three classical DOF. By combining all three DOF together, we obtain $2 \times 2 \times 3 = 12$ modes in total.

To conclude, the work is interesting but not presented clearly, and some imprecisions are made. In consequence, I can't recommend the publication in Nature Communications.

We thank the referee for those helpful comments which substantially improved the presentation of our experiments and motivated several new and interesting experiments. We hope that the Referee will reconsider his/her views in the light of our new experiments and revised manuscript.

Response to Reviewer #3 -- NCOMMS-17-29988

We thank the Referee for reading our manuscript carefully and for making useful suggestions and comments. We have carefully considered these comments and have revised the manuscript accordingly. We reply to the referee's comments below using blue font.

In the paper 'Multiplexed storage and real-time manipulation based on a multiple-degree-of-freedom quantum memory', the authors present a scheme in which photonic quantum information encoded in three different degrees of freedom can be stored and manipulated in a quantum memory.

The three different degrees of freedom that are used are the orbital angular momentum (OAM), the time and the frequency. The quantum memory protocol is the spin-wave atomic frequency comb protocol, which the authors use to

store coherent states at the single photon level with high signal-to-noise ratio. I consider that the developments that are proposed in this article are very timely, as densification of quantum information encoding and manipulation is a key resource for the development of large-scale quantum networks. However, two major points should in my opinion be addressed before I can make my final decision about the possibility to publish this work in Nature Communications.

We thank the Referee for these very positive comments.

- My first concern comes from the estimation of the fidelities between the input and manipulated output states, in the case of conversion or arbitrary manipulation. Indeed, all the fidelities that are presented in tables I and II in the article only concern the fidelity in the degree of freedom that is not manipulated (the OAM), which in my opinion is not relevant for characterizing the transformation. Other tools like process matrices in the time-frequency space (dimension 4 here) would be more adapted to prove that the transformation that is performed is indeed the one that is expected.

The quantum mode converter can transfer photonic pulses to a target temporal or spectral mode without distorting the photonic quantum states. The arbitrary manipulation of photonic pulses between different modes **while preserving photonic coherence** is an important requirement for many proposed photonic technologies [Nature 461, 241–245 (2009)]. In the case of conversion or arbitrary manipulation, the temporal and spectral DOF are employed as the classical DOF for multiplexing while the quantum information is encoded in the spatial DOF. To demonstrate that the qutrit state coherence is well preserved during conversion or arbitrary manipulation, we measured the fidelities between the input and manipulated states.

We carefully considered your helpful suggestion. The temporal DOF and spectral DOF here are simply employed as classical “channel” and the operations cannot be characterized using quantum process matrix. Nevertheless, we believe the “mode crosstalk” we presented in the revised manuscript can be a useful indicator for characterization of these operations.

- The second important point is related to the fidelity of the process matrices that the authors present for the OAM. It is very surprising to me to calculate the fidelity between two process matrices in this context (χ_{input} and χ_{output} , or χ_{ideal} and χ_{output}), given that we want to characterize the process associated to the quantum memory only. Indeed, if I understood correctly, χ_{output} is the process matrix calculated with the output density matrices of the whole process (preparation of the OAM and memory) and the input ideal density matrices (pure states), whereas it should be calculated using the density matrices in input of the quantum memory (mixed states, taking into

account the imperfect preparation). The relevant number would then be the 'identity component' of this process matrix.

Instead of this, the authors compare how similar are the process matrices with and without the quantum memory by calculating a fidelity between them. This fidelity, in my opinion, is not equivalent and less relevant than the previously mentioned 'identity component'.

We thank the Referee for this very helpful suggestion. According to Referee's suggestion, we have reconstructed the process matrix of our quantum memory by using the density matrices of the input of the quantum memory (mixed states, taking into account the imperfect preparation). In this condition, the memory fidelity is simply the 'identity component' of this process matrix. The result is presented in Fig. 2b.

In addition to these two major issues, a few points also raised questions during my reviewing process:

- Following the second point mentioned above, two numbers are in my opinion missing in the text: what are the fidelities of the prepared OAM qutrit states with the ideal ψ_1 and ψ_2 states? And what is limiting in this case? For instance, on figure S4 it looks like the $|G\rangle\langle G|$ component is higher than all the others: is it due to preparation imperfection or to the action of the memory?

The fidelities of the prepared OAM qutrit states with the ideal ψ_1 and ψ_2 states are 0.903 ± 0.004 and 0.895 ± 0.008 . We have presented these results in Supplementary Information section III. We believe the limiting factors are the imperfection in preparation, transmission and detection. Higher fidelity may be achieved by employing a better SLM and detection based on OAM mode sorter.

The $|G\rangle\langle G|$ component is higher than all the others. This is because of the imperfection of the setup and the action of the memory. Due to the imperfection in preparation and detection, the $|G\rangle\langle G|$ component is higher than other LG modes before memory. The storage process will further increase this deference due to the limited diameter of pump/control beam. We have included the discussion on this point in Supplementary Information section III.

- On figure 2b: could the authors precise what the λ_i operators are (even in the suppl. mat.), and give an intuition why the imperfections seem to mainly come from λ_4 and λ_5 ?

These operators are the complete operators in three dimension Hilbert space [Phys. Rev. A 66, 012303 (2002), Phys. Rev. Lett. 115, 070502 (2015)]. There operators are given as follows:

$$\lambda_1 = \begin{bmatrix} 1 & 0 & 0 \\ 0 & 1 & 0 \\ 0 & 0 & 1 \end{bmatrix}; \lambda_2 = \begin{bmatrix} 0 & 1 & 0 \\ 1 & 0 & 0 \\ 0 & 0 & 0 \end{bmatrix}; \lambda_3 = \begin{bmatrix} 0 & i & 0 \\ -i & 0 & 0 \\ 0 & 0 & 0 \end{bmatrix}; \lambda_4 = \begin{bmatrix} 1 & 0 & 0 \\ 0 & -1 & 0 \\ 0 & 0 & 0 \end{bmatrix}; \lambda_5 = \begin{bmatrix} 0 & 0 & 1 \\ 0 & 0 & 0 \\ 1 & 0 & 0 \end{bmatrix}; \lambda_6 = \begin{bmatrix} 0 & 0 & i \\ 0 & 0 & 0 \\ i & 0 & 0 \end{bmatrix}; \lambda_7 = \begin{bmatrix} 0 & 0 & 0 \\ 0 & 0 & 1 \\ 0 & 1 & 0 \end{bmatrix}; \lambda_8 = \begin{bmatrix} 0 & 0 & 0 \\ 0 & 0 & -i \\ 0 & i & 0 \end{bmatrix}; \lambda_9 = \frac{1}{\sqrt{3}} \begin{bmatrix} 1 & 0 & 0 \\ 0 & 1 & 0 \\ 0 & 0 & -2 \end{bmatrix}.$$

We have presented these operators in Methods in the revised manuscript.

Following your suggestion, now we reconstruct the memory process using the density matrices of the input of the quantum memory and the imperfections are no longer dominated by λ_4 and λ_5 .

- Could the authors explain why they chose a spin duration of 7.68 μs ? Has this particular number been chosen for noise issues?

The spin duration is an arbitrary choice. In our experiment, we used an AWG with 2.5G samples/s and generated a waveform with length of 19200 points which gives the spin duration of 7.68 μs . This number is not considered for noise issues.

- The combs that are presented in the paper allow to reach efficiencies which are 0.5% lower than in the case of a unique comb (section I of supl. mat.): is there a particular reason for this small decrease in the efficiency? Is the inhomogeneous profile responsible for this small drop (lower optical depth in order to match both efficiencies)? Also, regarding the combs: the authors use two temporal modes in combs that possess ten teeth. This means that overall, approximately ten modes could be used. Given that the SNR is high (almost 40 in their case), why did the authors limit themselves to two modes, and how dramatic is the decrease of the SNR if this number of modes is increased?

In order to achieve the temporal multiplexing, the duration of input pulses needs to be reduced. The duration of each input pulse is reduced from 500ns to 390ns, leading to a reduction of memory efficiency due to the larger bandwidth of input pulses. We have added this explanation in Supplementary Information section I.

As the Referee says, maximally ten modes could be used in these AFC. But increasing the number of modes, the time interval between the last control pulse and the first output signal pulse will be reduced. This will lead to increased noise and we limit our experiment to two modes. We have added this explanation in the 2nd paragraph on page 3 of the revised manuscript.

This limit can be overcome with long AFC echo times [Phys. Rev. A 93,032327 (2016)]. If the AFC echo time is long enough, the SNR will not increase with the

number of modes.

- The authors claim that two fidelities in table 2 are lower, due to 'less photon counts in each output'. But a decrease in the photon counts should decrease the precision and not the fidelity: here the fidelity is decreased, well below the error bars that are presented. Could the authors comment on this?

Temporal beam splitting resulted into less than 50% of the output signal in each temporal output mode as compared with other operations. Meanwhile, the noise in each output mode is the same. Therefore, the SNR for each temporal output mode should decrease and results into a lower fidelity.

- A 'classical benchmark' is mentioned regarding the states manipulation, before the final discussion. The authors claim to be well above it: could they precise which limit is mentioned here? More precisely, as the states that are used are weak coherent states, the limit fidelity of $2/3$ should strictly speaking not be used and a more complete criterion must be chosen.

We thank the Referee for the helpful suggestions. To demonstrate the quantum behavior of our memory, the measured fidelity is compared with the highest fidelity achievable with a measure-and-prepare approach, taking into account the Poissonian statistics of the input states and the finite memory efficiency [Nature 489, 541 (2012), Nature Photonics 8, 234 (2014), PRL 108, 190504 (2015)]. We have presented these results in Supplementary Information section IV. These results demonstrate the quantum nature of our memory.

Eventually, I also found some typos in the article:

- In the discussion about fidelities between process matrices, (χ_{ideal}) should be next to χ_{input} and not χ_{output} , and 'resp.' could be used to clarify the two fidelities that are presented.
- In the 'Arbitrary manipulation in real time', I think that the 'exchange of the readout times for the f_1 and f_2 photons' should read $f_1 t_2, f_2 t_1$ both in the text and in table II, as correctly written on figure 4.
- The z axes in figure S4 are misleading, as χ is usually used for process matrices, and ρ_{out} is the name that is mentioned in the methods.
- First paragraph, line 12: 'use' should be 'uses'
- Fourth paragraph, line 7: 'In addition to the increasing' should be 'In addition to increasing'
- Methods, line 3: 'Refs.' should be 'Ref.'
- [27]: 'Parigi1' should be 'Parigi'
- Caption of figure S4: a parenthesis is missing for state ψ_1 .

We sincerely thank the Referee for reading our manuscript carefully. We have

carefully checked the manuscript and corrected all the typos accordingly.

REVIEWERS' COMMENTS:

Reviewer #1 (Remarks to the Author):

I am satisfied with the authors' replies to my comments. The authors have adequately addressed the concerns I raised by making substantial changes to the manuscript.

Reviewer #2 (Remarks to the Author):

The authors have performed changes according to the referees' comments to improve the clarity of the manuscript. I still have some remarks, detailed below.

1. The total memory efficiency should appear in the main text (and then the details in the supplementary sections).
2. It's good that the authors clarified their definition of the SNR. It is in fact a measurement of the crosstalks, so I think the authors should call it simply crosstalk to avoid confusion for the readers.
3. About the number of modes, the authors replied that "To demonstrate the ability of multiplexed storage over 3 classical DOF, three independent spatial modes are involved in the revised experiment. The experimental results are presented in Fig. 3b and Fig. 3c." I don't see the difference between Fig 3c and Fig 3b of the previous version. I think they didn't (or didn't want to) get my point. The problem is that they represent a density matrix while there is no quantum state (the spectral and temporal DOF are classical modes).
4. Fig 5: Does the little "x 2" mean there is the same for the f2 frequency? For more clarity, they should had the two frequencies on top of each other (like for the 2nd graph from the top).
5. It is good that the authors calculated the classical bound for the fidelity. However, they should give the classical bound in the main text on p. 3, first column, just after giving the process fidelity (and not later in the second column as they do now). This will allow the reader to compare both number and be convinced about the quantum character of the memory. Moreover, in the supplementary sections, I don't understand why they give so many different formulas for the classical bounds, because they don't comment on them. It really lacks a discussion about the different models, and which one applies in this case. They should also add their experimental point on Fig S4.

Reviewer #3 (Remarks to the Author):

After reading the revised version of the paper, I consider that my concerns are now properly addressed.

Indeed, a detailed point-by-point answer of my comments has been done with care, and I can now meet my final decision of acceptance of the paper for publication in Nature Communications.